# Traveler's knowledge, attitude, and practice about travel health insurance: A community-based questionnaire study

**Chia-Jung Yang**[1,2], **Chia-Wen Lu**[3,4], **Chien-Hsieh Chiang**[3,4,5], **Hao-Hsiang Chang**[3,4], **Chien-An Yao**[3,4], **Kuo-Chin Huang**[2,3,4,6]*

1 Department of Family Medicine, Kinmen Hospital, Ministry of Health and Welfare, Kinmen, Taiwan,
2 Department of Family Medicine, National Taiwan University Hospital Bei-Hu Branch, Taipei, Taiwan,
3 Department of Family Medicine, National Taiwan University Hospital, Taipei, Taiwan, 4 Department of Family Medicine, National Taiwan University College of Medicine, Taipei, Taiwan, 5 Department of Community and Family Medicine, National Taiwan University Hospital Yunlin Branch, Yunlin, Taiwan, 6 Department of Family Medicine, National Taiwan University Hospital Hsinchu Branch, Hsinchu, Taiwan

* bretthuang@ntu.edu.tw

## Abstract

### Background

Travel, especially international travel, has become one of the most popular leisure activities in the world. The risk of accidents and travel-related illnesses, including infectious and non-communicable diseases, should not be neglected. To provide a more comprehensive pre-travel consultation to international travelers, this study aimed to investigate the knowledge, attitude, and practice of travelers about travel health insurance.

### Methods

This was a cross-sectional study. Anonymous structured questionnaires were distributed to 1000 visitors to the Taiwan International Travel Fair in May 2019.

### Results

The top three important travel health insurances were accidental death and disablement insurance (92%), accidental medical reimbursement (90.4%), and 24-hour emergency assistance (89%). In addition to education level, travel-associated illness, and special activities during travel, a significant association was observed between the willingness to buy various travel health insurances and the willingness of pre-travel consultation.

### Conclusions

Most travelers would buy travel health insurance; however, disproportional respondents understood the content of travel health insurance. Most travelers considered travel clinics to be the most reliable information source regarding travel health insurance. Therefore, travel

**Data Availability Statement:** All relevant data are within the paper and its Supporting information files.

**Funding:** This study has been partially sponsored by the Centers for Disease Control, Taiwan (JK108026). The funders had no role in study design, data collection and analysis, decision to publish, or preparation of the manuscript.

**Competing interests:** The authors have declared that no competing interests exist.

medicine specialists are encouraged to offer more information about travel health insurance during pre-travel consultation.

## Introduction

International travel is appealing owing to globalization and the increasing trend of international tourism. According to the World Tourism Organization, the number of international arrivals was 1.442 billion in 2018 [1]. As international tourism increases and travel destinations diversify, more travelers acquire infectious diseases, which are not endemic in their home countries [2]. Up to 6%–87% of travelers became ill during or after their travels [3]. Keeping travelers healthy is the health providers' mission and the responsibility is shared with travelers and health providers. However, 80% of European travelers were found to be non-compliant with the traditionally recommended dietary restriction, and 20% of all travelers did not carry any antimalarial drugs [4]. Most of the travelers did not follow the suggestions even though they received suggestions about disease prevention from their healthcare providers. Besides, accidents happen even when one is well-prepared. Approximately 20%–25% of travelers' deaths were caused by injuries, and road traffic injuries were its leading cause [5]. Travel insurance is one of the most important safety nets for travelers and should be reinforced by travel health advisers [6].

Recent studies have shown that approximately 60% of General Practitioners in New Zealand [7] and 39% of travel clinics worldwide [8] usually advise travelers to buy travel insurance. Furthermore, as our previous study has shown, overseas emergency medical assistance services (EMAS) were considered important to travelers, but approximately 20%–30% of travelers lacked awareness about EMAS [9]. Since travel medicine services have also been indispensable in the COVID-19 pandemic, we realized that pre-travel healthcare will be even more vital in the future [10]. However, there is limited evidence regarding the willingness and the awareness of travelers about travel health insurance. Moreover, the travelers' sources of buying travel health insurance and the correlation between the willingness to seek pre-travel consultation and the willingness to buy travel health insurance have not been well investigated. Therefore, this study was conducted to investigate the travelers' knowledge, attitude, practice, and sources about travel health insurance to help healthcare workers provide more comprehensive pre-travel consultation to international travelers.

## Materials and methods

### Design

This study was a community-based, cross-sectional questionnaire survey. The questionnaire was self-administered and anonymous. The study was approved by the Institutional Review Board at National Taiwan University Hospital in Taiwan (201902070W) before the study was conducted.

### Subjects

This survey included visitors who attended the Taiwan International Travel Fair in May 2019. Inclusion criteria were as follows: aged >20 years, willing, and able to complete the questionnaire. All respondents gave verbal consent before they fulfilled the questionnaire.

## Questionnaire

The four-part questionnaire included questions on socio-demographical characteristics, knowledge of travel-related diseases and vaccines, attitudes, awareness, and willingness toward visiting travel medicine clinics and buying travel insurance. The questionnaire was pretested for face validity by a committee of ten physicians. The members of the committee were from National Taiwan University Hospital (NTUH) and Centers for Disease Control (CDC), Taiwan who were experienced in the clinical practice of travel medicine. Literature review was conducted and consensus opinion from three physicians at NTUH and CDC, Taiwan, was taken for testing the content validity of the questionnaire.

The socio-demographical characteristics included sex, age, education level, occupation, medical history, and special activities during travel. The other three parts of the questionnaire included the following components:

The knowledge of travel-related disease: 10 questions about vaccination, malaria, yellow fever, cholera, measles, hepatitis B, rabies, meningococcus, and influenza. Each question was scored 1 point for a correct response, with a total score of 10 points. These questions tested the respondents' knowledge regarding the epidemiology, medication, and vaccination of travel-related disease.

Attitude toward travel medicine clinics and travel health insurance: This part examined the participants' perceptions regarding to the importance of travel medicine clinics and travel health insurance. It included questions regarding: (1) travel medicine clinics, (2) pre-travel vaccination, (3) accidental death and disablement insurance, (4) accidental medical reimbursement, (5) overseas sickness coverage, (6) 24-hour emergency assistance, and (7) travel inconvenience insurance. The scoring system used a five-point Likert Scale, ranging from "very unimportant" (1 point), "unimportant" (2 points), "no comment" (3 points), "important" (4 points), to "very important" (5 points). Higher scores indicated positive attitudes regarding the need of certain services.

Awareness and willingness toward visiting travel medicine clinics and buying travel health insurance: This part sought the information on whether the participants heard about the abovementioned five different kinds of travel health insurances and travel clinics and whether or not they will use the service.

## Statistical analysis

Data were presented as mean ± SD for continuous variables and numbers (%) for categorical variables. The chi-square test was used to compare the proportion of the willingness to buy travel insurance between different socio-demographic variables. One-way analysis of variance and independent t-test were used to clarify the relation between the knowledge of travel medicine and socio-demographic variables. A $p$-value $<0.05$ was considered statistically significant. Statistical analysis was carried out using the statistical software, Statistical Package for the Social Sciences 19.0 (version: 19.0, IBM Corp., Armonk, NY, USA, 2017).

## Results

A total of 1,000 participants were randomly administered the questionnaire, and 927 visitors responded (response rate = 92.7%) to it. The high response rate and unbiased selections represented good internal validity. After eliminating 99 incomplete questionnaires, the final analysis included 828 respondents (303 males and 525 females).

Table 1 shows the demographic characteristics of the respondents. The mean age of the respondents was 44.7 ± 14.0 years. Among the respondents, 32.1% (n = 266) reported a medical history of chronic illnesses and 19.2% (n = 159) reported a previous experience of travel-

**Table 1. Basic characteristics of survey respondents (N = 828).**

| Characteristic | Number | Percentage (%) |
|---|---|---|
| Age (mean = 44.7±14.0 years) | | |
| 20–29 | 127 | 15.3 |
| 30–39 | 202 | 24.4 |
| 40–49 | 209 | 25.2 |
| 50–69 | 149 | 18 |
| 60–89 | 141 | 17 |
| Sex | | |
| Female | 525 | 63.4 |
| Male | 303 | 36.6 |
| Education level | | |
| High school or below | 172 | 20.8 |
| University or college | 532 | 64.2 |
| Graduate school and higher | 124 | 15 |
| Medical history [a] | | |
| No | 562 | 67.9 |
| Yes | 266 | 32.1 |
| Purpose of trip (multiple choices) | | |
| Self-guide tour | 446 | 53.9 |
| Group tours | 318 | 38.4 |
| Business travel | 32 | 3.9 |
| Visiting friend and relatives | 21 | 2.5 |
| Study abroad | 7 | 0.8 |
| Live abroad | 7 | 0.8 |
| International volunteering | 4 | 0.5 |
| Travel-associated illness [b] | | |
| No | 669 | 80.8 |
| Yes | 159 | 19.2 |
| Planned special activities [c] | | |
| No | 645 | 77.9 |
| Yes | 183 | 22.1 |

[a] Hypertension, diabetes, hyperlipidemia, cardiovascular disease, gout, chronic renal disease, liver disease, chronic urticaria, thyroid disease, vasculitis, sleep disorder

[b] Common cold, influenza, travelers' diarrhea, trauma, herpes zoster, conjunctivitis, animal bite

[c] Mountain backpacking, scuba diving, river rafting, snow skiing, surfing, marathon, cycling, glacier hiking

associated illness. Additionally, 38.4% (n = 318) of the respondents planned to join group tours and 53.9% (n = 446) planned a self-guided tour. Furthermore, a total of 22.1% (n = 183) of the respondents planned to participate in activities that may increase their risk of health problems, such as mountain backpacking or scuba diving, during their travel.

Table 2 shows the association between the willingness to buy different types of travel health insurances and demographic characteristics. A statistically significant association was observed between the willingness to buy travel inconvenience insurance and educational level (p-value = 0.001). The willingness of pre-travel consultation was associated with various travel health insurances (p-value < 0.001), and travel-associated illness was associated with the willingness to buy a 24-hour emergency assistance (p-value = 0.031). Furthermore, planned

**Table 2. Association between various characteristics and the willingness to buy different travel insurances (N = 828).**

| Variable | Number | Accidental death and disablement insurance | | | Accidental medical reimbursement | | | Overseas sickness coverage | | | 24-hour of emergency assistance | | | Travel inconvenience insurance | | |
|---|---|---|---|---|---|---|---|---|---|---|---|---|---|---|---|---|
| | | Unwilling N (%) | Willing N (%) | P Value | Unwilling N (%) | Willing N (%) | P Value | Unwilling N (%) | Willing N (%) | P Value | Unwilling N (%) | Willing N (%) | P Value | Unwilling N (%) | Willing N (%) | P Value |
| **Age(years)** | | | | 0.748 | | | 0.311 | | | 0.346 | | | 0.415 | | | 0.097 |
| 20–29 | 127 | 1.6 | 78.7 | | 7.1 | 73.2 | | 7.1 | 66.1 | | 7.9 | 64.6 | | 9.4 | 70.1 | |
| 30–39 | 202 | 2 | 77.2 | | 3 | 75.7 | | 5.9 | 66.8 | | 7.9 | 66.3 | | 5.4 | 74.8 | |
| 40–49 | 209 | 1.9 | 77 | | 1.4 | 77.5 | | 4.3 | 67.9 | | 2.9 | 73.2 | | 6.7 | 67.5 | |
| 50–59 | 149 | 3.4 | 81.2 | | 4 | 77.9 | | 6.7 | 69.1 | | 7.4 | 68.5 | | 8.7 | 73.8 | |
| 60–89 | 141 | 4.3 | 75.9 | | 5.7 | 76.6 | | 9.9 | 72.3 | | 7.8 | 70.9 | | 14.2 | 66 | |
| **Sex** | | | | 0.78 | | | 0.276 | | | 0.579 | | | 0.53 | | | 0.667 |
| Female | 525 | 2.5 | 78.7 | | 3.2 | 77.9 | | 6.9 | 69.1 | | 6.7 | 60.1 | | 7.8 | 71.2 | |
| Male | 303 | 2.6 | 76.6 | | 5 | 73.6 | | 5.9 | 67 | | 6.3 | 67 | | 9.6 | 69.3 | |
| **Educational level** | | | | 0.115 | | | 0.077 | | | 0.085 | | | 0.091 | | | 0.001** |
| High School or below | 172 | 4.7 | 73.3 | | 6.4 | 70.3 | | 8.1 | 63.4 | | 7.6 | 65.7 | | 14.5 | 62.8 | |
| University or College | 532 | 2.3 | 78 | | 3.6 | 76.7 | | 6.8 | 67.7 | | 7.3 | 68 | | 7.5 | 70.1 | |
| Graduate School and higher | 124 | 0.8 | 83.9 | | 1.6 | 83.1 | | 3.2 | 78.2 | | 1.6 | 77.4 | | 4 | 83.1 | |
| **Medical history** | | | | 0.271 | | | 0.087 | | | 0.518 | | | 0.785 | | | 0.195 |
| No | 562 | 2 | 78.8 | | 2.8 | 77.2 | | 5.9 | 68.5 | | 6.2 | 68.7 | | 7.3 | 71.9 | |
| Yes | 266 | 3.8 | 75.9 | | 6 | 74.4 | | 7.9 | 68 | | 7.1 | 69.5 | | 10.9 | 67.7 | |
| **Willingness of pre-travel consultation** | | | | 0.000** | | | 0.000** | | | 0.000** | | | 0.000** | | | 0.000** |
| Yes | 493 | 1.8 | 86.8 | | 3.4 | 85.4 | | 5.7 | 79.7 | | 4.5 | 81.3 | | 6.7 | 79.5 | |
| Undetermined | 307 | 2 | 64.5 | | 3.3 | 62.9 | | 6.8 | 50.5 | | 7.8 | 50.8 | | 9.1 | 58 | |
| No | 28 | 21.4 | 67.9 | | 17.9 | 64.3 | | 17.9 | 64.3 | | 28.6 | 50 | | 32.1 | 50 | |
| **Travel-associated illness** | | | | 0.082 | | | 0.235 | | | 0.182 | | | 0.031*+ | | | 0.249 |
| No | 669 | 2.2 | 76.8 | | 3.4 | 75.9 | | 6.3 | 67.3 | | 6 | 67.7 | | 7.9 | 70.1 | |
| Yes | 159 | 3.8 | 82.4 | | 5.7 | 78 | | 7.5 | 73 | | 8.8 | 74.2 | | 10.7 | 72.3 | |
| **Planned special activities during travel** | | | | 0.006** | | | 0.068 | | | 0.076 | | | 0.043* | | | 0.307 |
| No | 645 | 1.7 | 77.5 | | 3.3 | 75.7 | | 5.7 | 67.8 | | 6.4 | 67.1 | | 7.8 | 70.5 | |
| Yes | 183 | 5.5 | 79.2 | | 6 | 78.7 | | 9.3 | 70.5 | | 7.1 | 75.4 | | 10.9 | 70.5 | |

The chi-square test was applied to compare the proportion of the willingness to buy travel insurances between different socio-economic variables.

*p value < 0.05;

**p value < 0.001

**Table 3. Comparison of mean scores of knowledge of travel medicine with socio-demographic variables.**

| Variable | Number | Mean(±SD) | P value |
|---|---|---|---|
| Age(years) | | | 0.000** |
| 20–29 | 127 | 5.6(±1.68) | |
| 30–39 | 202 | 6.13(±1.75) | |
| 40–49 | 209 | 6.68(±1.85) | |
| 50–59 | 149 | 6.81(±2.02) | |
| 60–89 | 141 | 7.5(±1.94) | |
| Medical history | | | 0.000** |
| No | 562 | 6.38(±1.87) | |
| Yes | 266 | 6.89(±2.0) | |
| Travel-associated illness | | | 0.035* |
| No | 669 | 6.48(±1.91) | |
| Yes | 159 | 6.83(±2.0) | |
| Planned special activities during travel | | | 0.532 |
| No | 645 | 6.52(±1.9) | |
| Yes | 183 | 6.62(±2.1) | |

One-way analysis of variance and independent t-test were applied to clarify the relation between the knowledge of travel medicine and socio-demographic variables.

*p value < 0.05;

**p value < 0.001

special activities during travel were associated with accidental death (p-value = 0.006) and disablement insurance (p-value = 0.043).

Table 3 shows the comparison of the mean scores of the knowledge of travel medicine with socio-demographic variables. The knowledge of travel-related infection, vaccine, and diseases was significantly associated with age, medical history, and travel-associated illness.

Table 4 shows the source and subjective trust of travel health insurance information. Most respondents obtained their information from social media (45.9%), insurance company (45.4%) or travel agency (35.9%). However, the majority of respondents appraised healthcare workers (44.1%), and insurance company (42.5%) as the most reliable sources.

## Discussion

In our main findings, we clearly demonstrated that travelers when asked about the most important insurances or to rate the insurances, 92% answered accidental death and disablement insurance, 90.4% answered accidental medical reimbursement, and 89% answered 24-hour emergency assistance. Risk management is an important issue for travelers, since accidents and injuries are important causes of morbidity and mortality [11, 12]. Buying travel

**Table 4. Source and subjective trust of travel health insurance information (N = 828).**

| Variable | Source | Subjective trust |
|---|---|---|
| Social media | 45.9% | 31.3% |
| Insurance company | 45.4% | 42.5% |
| Travel agency | 35.9% | 30.1% |
| Family or friends | 24.2% | 18.8% |
| Healthcare workers | 10.7% | 44.1% |

health insurance is one of the methods to reduce the risk of loss. There were few studies investigating travel health insurance, and most studies were conducted from the health advisor's perspective [7, 8]. Our study focused on another viewpoint from the travelers.

We found that the percentage of willingness to buy travel health insurance was high, about 60%–80% in this study. However, previous study showed that most travelers tend to consider the price instead of the service details and the quality coverage when purchasing travel insurance [13]. Besides, the low percentage of travelers (59.5%) willing to seek pre-travel advice described in the literature contrasts with the percentage (62.8–86.8%) of those motivated to buy travel health insurance in this study. In another view, it's a good opportunity to raise their awareness of pretravel consultation when travelers purchase travel health insurance. If the two disciplines cooperate, travelers can receive more comprehensive health care.

Our study revealed only 31.5%–44.7% of respondents understood the content of travel insurance very well, whereas 9.5%–17.3% expressed their ignorance. The previous study showed only two-thirds of travel insurance claims were fully approved, and most refusal was due to pre-existing conditions and poor documentation [14]. During the COVID-19 pandemic, many insurance companies discontinued selling travel health insurance, or any coverage related to COVID-19 [10]. Therefore, travelers must be aware of the health insurance policy, especially the coverage when infected with newly emerging infectious diseases or pre-existing condition while abroad [15, 16].

Additionally, we found that fewer respondents were willing to buy overseas sickness coverage (68.4%) and 24-hour emergency assistance (69%). However, emergency medical evacuation from a low-income nation costs $50,000 to $75,000 or more [17]. Although less than 0.5% of travelers need medical evacuation, there were still considerable cases in the booming generation of tourism [18]. According to the statistics of International SOS Group of Companies, they received 4 million assistance calls until 2022 [19]. The demand of 24-hour emergency assistance is quite a lot. Overseas sickness coverage and medical evacuation are crucial while traveling during the Covid-19 pandemic since local healthcare capacity can be overwhelmed [20].

Travelers with high educational levels, willingness to pre-travel consultation, travel-associated illnesses, and planned special activities during travel were more willing to buy travel health insurance. In addition, travelers with older age, medical histories, and travel-associated illnesses were significantly associated with a higher knowledge of travel medicine. In other words, those travelers who have the willingness to buy travel health insurance may not have good knowledge of travel medicine. Compared with a previous study, approximately 40% of European travelers could not assess the risk of infectious diseases accurately [21]. Buying travel health insurance without adequate consultation is not enough for travel health and safety. Strengthening travelers' awareness about travel-related diseases is also important.

It is not surprising that most respondents reported that the source of their travel-related and insurance knowledge was social media or insurance company, but they thought that healthcare workers were the most reliable source. However, little nonmedical health advice was offered to travelers by healthcare workers in general practice. Previous study has also shown that the variance in medical coverage of different travel health insurance is complex [14]. Therefore, healthcare workers are encouraged to be equipped with the knowledge of travel health insurance to help travelers.

## Limitations

This study has some limitations. First, the questionnaire was conducted in Taiwan International Travel Fair in north Taiwan and might result in sampling bias. Most of the respondents

were younger, with high education levels, with less experience of travel-associated illness, and with less pre-existing diseases; thus, our findings might not be applicable to all travelers. Nevertheless, few studies have investigated travelers' attitude, willingness, and awareness about travel health insurance. We believe that this study will give an insight into pre-travel health consultation in the future.

## Conclusions

Between a third and half of the travelers understand the content of health insurance, almost all recognize its importance, and more than two-thirds were willing to buy a health insurance when traveling. However, just buying insurance does not equal to being well-prepared. Understanding local epidemic and implement personal protective measures is still important. Most travelers thought that healthcare workers are the most reliable source of travel insurance. Therefore, health professionals should be encouraged to acquire health insurance knowledge and transmitted it on during pre-travel consultations before the countries reopen for tourism.

## Supporting information

**S1 File. Minimal data set.**
(XLSX)

## Acknowledgments

We would like to thank Miss Chia-Chi Yu for her assistance in this study.

## Author Contributions

**Data curation:** Chia-Wen Lu.

**Formal analysis:** Chia-Jung Yang, Chia-Wen Lu.

**Investigation:** Chia-Wen Lu.

**Methodology:** Chia-Wen Lu.

**Project administration:** Chia-Jung Yang.

**Resources:** Kuo-Chin Huang.

**Supervision:** Chia-Wen Lu, Chien-Hsieh Chiang, Kuo-Chin Huang.

**Writing – original draft:** Chia-Jung Yang.

**Writing – review & editing:** Chia-Wen Lu, Chien-Hsieh Chiang, Hao-Hsiang Chang, Chien-An Yao, Kuo-Chin Huang.

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
