## [Decision Letter · Decision Letter 0]

29 Jun 2022

PONE-D-21-34714

Traveler’s Knowledge, Attitude, and Awareness about Travel Health Insurance during the COVID-19 Pandemic

PLOS ONE

Dear Dr. Huang,

Thank you for submitting your manuscript to PLOS ONE. Firstly, we would like to apologize for the delay in processing your manuscript. It has been exceptionally difficult to secure reviewers to evaluate your study. We have now received one completed review, which is available below.

Please note that we have only been able to secure a single reviewer to assess your manuscript. We are issuing a decision on your manuscript at this point to prevent further delays in the evaluation of your manuscript. Please be aware that the editor who handles your revised manuscript might find it necessary to invite additional reviewers to assess this work once the revised manuscript is submitted. However, we will aim to proceed on the basis of this single review if possible.

After careful consideration, we feel that it has merit but does not fully meet PLOS ONE’s publication criteria as it currently stands. Therefore, we invite you to submit a revised version of the manuscript that addresses the points raised during the review process.

We look forward to receiving your revised manuscript.

Kind regards,

Miquel Vall-llosera Camps

Senior Editor

PLOS ONE

“This study has been partially sponsored by the Centers for Disease Control, Taiwan

(JK108026).”

:

Additional Editor Comments:

We noticed that your manuscript title and abstract specifically mentions COVID-19. However, the survey was conducted in March 2019, just at the beginning of the global COVID-19 pandemic, and we assessed that the article was written to represent a much broader subject of international travel unrelated to COVID-19. We would therefore suggest removing the mention of COVID-19 from the title and abstract.

Reviewers' comments:

Reviewer's Responses to Questions

**Comments to the Author**

1. Is the manuscript technically sound, and do the data support the conclusions?

Reviewer #1: Yes

2. Has the statistical analysis been performed appropriately and rigorously? 

Reviewer #1: Yes

3. Have the authors made all data underlying the findings in their manuscript fully available?

Reviewer #1: Yes

4. Is the manuscript presented in an intelligible fashion and written in standard English?

Reviewer #1: Yes

5. Review Comments to the Author

Reviewer #1: A few things to consider in revisions:

1. This study is a knowledge, attitudes, and awareness study, but I am curious as to why "practices" was not included? To me, awareness is the same as knowledge, so I think this important component was left out of the questionnaire.

2. I am not sure what is meant by "important travel insurances." Typically travel insurance has multiple facets, and one needs to look to see what coverage is.

3. Remove the study from the 1980s from the intro. It is old and not relevant.

4. International SOS is not mentioned at all in the manuscript. As the largest provider of insurance and evacuation services globally they should be mentioned. They have also published findings they do not seem to be included.

5. Were respondents consented? If yes or no, this should be mentioned in the methods.

6. One third of people had a chronic illness, which likely greatly impacted their responses. This should be discussed thoroughly in the discussion.

7. All tables should be in descending numerical order. (example: purpose of trip is not in any sort of format). Right now each variable is not in descending order.

8. The discussion should start with a paragraph on what makes this analysis important and unique.

9. I am unclear what this means in the context of the discussion: "An increasing trend for international travel has been observed." Remove.

10. Conclusions should not have any data. They should reiterate the take home points without data. I also think the conclusions need to be strengthened. What is the public health and clinical importance of the findings?

11. Each paragraph of the discussion should be expanded on. Paragraphs should start with a strong introductory sentence, available evidence, and then followed by a strong closing.

12. I do not think that Figure 1 adds much. Remove.

6. PLOS authors have the option to publish the peer review history of their article (what does this mean?). If published, this will include your full peer review and any attached files.

Reviewer #1: No

---

## [Author Response · Author response to Decision Letter 0]

2 Aug 2022

Replay to Edits (2022/8/2)

We've checked your submission and before we can proceed, we need you to address the following issues:

1. Please provide additional details regarding participant consent. In the Methods section, please ensure that you have specified (1) whether consent was informed and (2) what type you obtained (for instance, written or verbal). If your study included minors, state whether you obtained consent from parents or guardians. If the need for consent was waived by the ethics committee, please include this information.

Response: Thanks for your recommendation. We stated that the purpose of this study was to understand travelers’ knowledge, attitudes, and practice at the beginning of the questionnaire. And all respondents gave verbal consent before they fulfilled the questionnaire. We have also shown “The study was approved by the Institutional Review Board at National Taiwan University Hospital in Taiwan (201902070W) before the study was conducted.” on page 7. The questionnaire was administered anonymously in public. We cannot recognize the personal identification after completion of the questionnaire. The Institutional Review Board at National Taiwan University Hospital in Taiwan agreed that the study could be conducted with oral informed consent. 

Reply to Editor 

1. Please ensure that your manuscript meets PLOS ONE's style requirements, including those for file naming. The PLOS ONE style templates can be found at https://journals.plos.org/plosone/s/file?id=wjVg/PLOSOne_formatting_sample_main_body.pdf and https://journals.plos.org/plosone/s/file?id=ba62/PLOSOne_formatting_sample_title_authors_affiliations.pdf.

Response: Thanks for your recommendation. We’ve checked and amended our manuscript to meet PLOS ONE's style requirements. 

“This study has been partially sponsored by the Centers for Disease Control, Taiwan (JK108026).”Please state what role the funders took in the study. If the funders had no role, please state: "The funders had no role in study design, data collection and analysis, decision to publish, or preparation of the manuscript." If this statement is not correct you must amend it as needed. Please include this amended Role of Funder statement in your cover letter; we will change the online submission form on your behalf.

Response: Thanks for your recommendation. We’ve added "The funders had no role in study design, data collection and analysis, decision to publish, or preparation of the manuscript." in cover letter. 

Important: If there are ethical or legal restrictions to sharing your data publicly, please explain these restrictions in detail. Please see our guidelines for more information on what we consider unacceptable restrictions to publicly sharing data: http://journals.plos.org/plosone/s/data-availability#loc-unacceptable-data-access-restrictions. Note that it is not acceptable for the authors to be the sole named individuals responsible for ensuring data access. We will update your Data Availability statement to reflect the information you provide in your cover letter.

Response: Thanks for the instructions. We’ve uploaded minimal underlying data set as Supporting Information files, S1 File Minimal data set. 

Additional Editor Comments:

We noticed that your manuscript title and abstract specifically mentions COVID-19. However, the survey was conducted in March 2019, just at the beginning of the global COVID-19 pandemic, and we assessed that the article was written to represent a much broader subject of international travel unrelated to COVID-19. We would therefore suggest removing the mention of COVID-19 from the title and abstract.

Response: We appreciate your comment. We’d like to adjust the title to “Traveler’s Knowledge, Attitude, and Practice about Travel Health Insurance”. We also removed the mention of COVID-19 from the abstract and keywords. 

Reply to Reviewer

Reviewer #1

Reviewer #1: A few things to consider in revisions:

1. This study is a knowledge, attitudes, and awareness study, but I am curious as to why "practices" was not included? To me, awareness is the same as knowledge, so I think this important component was left out of the questionnaire.

Response: We appreciate your opinion. Actually, we included the travelers’ willingness toward visiting travel medicine clinics and buying travel health insurance, so we think that the revised title “Traveler’s Knowledge, Attitude, and Practice about Travel Health Insurance” is more appropriate. 

2. I am not sure what is meant by "important travel insurances." Typically travel insurance has multiple facets, and one needs to look to see what coverage is.

Response: Thanks for your comments. We feel the same way, the importance of the travel health insurance depends on the coverage and the demand of the traveler. Because we want to clarify the attitude and the willingness of the travelers, we asked the travelers how do they think about different kind of insurance and will they buy that. The result was according to their reply. 

3. Remove the study from the 1980s from the intro. It is old and not relevant.

Response: Thanks for your suggestion. We’ve removed the reference. 

4. International SOS is not mentioned at all in the manuscript. As the largest provider of insurance and evacuation services globally they should be mentioned. They have also published findings they do not seem to be included.

Response: Thanks for your suggestion. We agreed and used their statistics to show the need of 24-hour emergency assistance is quit a lot, so we added “According to the statistics of International SOS Group of Companies, they received 4 million assistance calls until 2022 [18]. The demand of 24-hour emergency assistance is quite a lot.” in discussion section, Line 226-227, Page 17. 

5. Were respondents consented? If yes or no, this should be mentioned in the methods.

Response: Thanks for your reminder. Yes. All respondents consented. We added that” All respondents gave verbal consent before they fulfilled the questionnaire.” in the Design paragraph, Line 109-110, Page 7.

6. One third of people had a chronic illness, which likely greatly impacted their responses. This should be discussed thoroughly in the discussion.

Response: Thanks for your recommendation. There was no significant association between chronic illness and the willingness to buy travel health insurance, thus we did not expound it (Table 2). However, we found that the knowledge of travel-related infection, vaccine, and diseases was significantly associated with medical history (Table 3). 

 We stated “In addition, travelers with older age, medical histories, and travel-associated illnesses were significantly associated with a higher knowledge of travel medicine. In other words, those travelers who have the willingness to buy travel health insurance may not have a good knowledge of travel medicine.” in discussion section, Line 233-237, Page 17. 

Table 2 Association between various characteristics and the willingness to buy different travel insurances. 

Table 3 Comparison of mean scores of knowledge of travel medicine with socio-demographic variables. 

7. All tables should be in descending numerical order. (example: purpose of trip is not in any sort of format). Right now each variable is not in descending order.

Response: Thanks for your recommendation. We’ve revised all tables, Table 1, 2 and 3. 

8. The discussion should start with a paragraph on what makes this analysis important and unique.

Response: We appreciate for your recommendation. We’ve now revised the first paragraph of Discussion section and focused on the unique of this study.” In our main findings, we clearly demonstrated that the top three important travel health insurances were accidental death and disablement insurance (92%), accidental medical reimbursement (90.4%), and 24-hour emergency assistance (89%). Furthermore, we found that the education level, travel-associated illness, and special activities during travel were significant parameters to influence the willingness to buy various travel health insurances and willingness of pre-travel consultation.” (Line 199-204, page 16).

9. I am unclear what this means in the context of the discussion: "An increasing trend for international travel has been observed." Remove.

Response: Thanks for your suggestion. We’ve removed it.

10. Conclusions should not have any data. They should reiterate the take home points without data. I also think the conclusions need to be strengthened. What is the public health and clinical importance of the findings?

Response: Thanks for your comments. We’ve revised the paragraphs of conclusion. Our study highlights the gap between travelers’ willingness to buy and to understand travel health insurance. Most travelers thought that healthcare workers are the most reliable source, so we should provide them more comprehensive advice, including travel health insurance.” Most travelers would buy travel health insurance, such as accidental death and disablement insurance and accidental medical reimbursement; however, a disproportionate percentage of the respondents understood the content very well. Just buying insurance does not equal to being well-prepared.” Line 261-264, Conclusions section, page 20. 

11. Each paragraph of the discussion should be expanded on. Paragraphs should start with a strong introductory sentence, available evidence, and then followed by a strong closing.

Response: Thanks for your comments. We’ve revised every paragraph of discussion according to your suggestion. Discussion section, page 16-18.

12. I do not think that Figure 1 adds much. Remove.

Response: Thanks for your suggestion. We’ve removed it.

---

## [Decision Letter · Decision Letter 1]

4 Oct 2022

PONE-D-21-34714R1Traveler’s Knowledge, Attitude, and  Practice about Travel Health InsurancePLOS ONE

Dear Dr. Huang,

Thank you for submitting your manuscript to PLOS ONE. After careful consideration, we feel that it has merit but does not fully meet PLOS ONE’s publication criteria as it currently stands. Therefore, we invite you to submit a revised version of the manuscript that addresses the points raised during the review process. Your manuscript has been evaluated by 4 reviewers and their comments are attached below. They complement you for your effort with the revisions, but raised a couple more concerns about the study and presentation. They suggest to include more discussion about what can be done to increase knowledge about travel insurance and would like to see inclusion of a discussion about the results presented in the tables. They also have a couple grammatical suggestions and queries about clarifications.Would you be able to address all their concerns for a resubmission of the manuscript?

We look forward to receiving your revised manuscript.

Kind regards,

Thomas Tischer

Staff Editor

PLOS ONE

Reviewers' comments:

Reviewer's Responses to Questions

**Comments to the Author**

1. If the authors have adequately addressed your comments raised in a previous round of review and you feel that this manuscript is now acceptable for publication, you may indicate that here to bypass the “Comments to the Author” section, enter your conflict of interest statement in the “Confidential to Editor” section, and submit your "Accept" recommendation.

Reviewer #1: All comments have been addressed

Reviewer #2: (No Response)

Reviewer #3: All comments have been addressed

Reviewer #4: All comments have been addressed

2. Is the manuscript technically sound, and do the data support the conclusions?

Reviewer #1: Yes

Reviewer #2: Yes

Reviewer #3: No

Reviewer #4: Yes

3. Has the statistical analysis been performed appropriately and rigorously? 

Reviewer #1: (No Response)

Reviewer #2: Yes

Reviewer #3: No

Reviewer #4: Yes

4. Have the authors made all data underlying the findings in their manuscript fully available?

Reviewer #1: (No Response)

Reviewer #2: Yes

Reviewer #3: Yes

Reviewer #4: Yes

5. Is the manuscript presented in an intelligible fashion and written in standard English?

Reviewer #1: Yes

Reviewer #2: Yes

Reviewer #3: No

Reviewer #4: Yes

6. Review Comments to the Author

Reviewer #1: (No Response)

Reviewer #2: I suggest you add the study’s design in the title and in the abstract.

In the abstract substitute:

Accident  accidents

visitors of  visitors to the

willingness  the willingness

the international  international

In the introduction substitute:

Increased  increasing

Tourisms  tourism

Responsibilities  responsibility

Noncompliant  non-compliant

1980s’  1980s

Shown  showed / has shown

Categories  the categories

About  of

Risk-coverage  risk coverage

a travel  travel

77 - 80 – you should clarify this sentence. First you say it is the health providers responsibility and then you say travellers did not follow the suggestions. Maybe the responsibility is shared?

93 - 95 – How the first part of the sentence justifies the second part?

In general, the introduction is not very clear and effective in demonstrating the importance of health insurance for travel and have several writing issues. I recommend that the text be proofread by a native English speaker.

111 – 114 - You should clarify this paragraph. It is not clear what is the study design, the instrument and the participants.

117 – 119 – This phrase does not belong in the subjects’ section.

In the subject’ section you should add from where and how the participants were selected.

Results

In the results’ section consider to include the number next to the percentage and also de p value next to the result presented.

The results presented in the tables should be better described in the text.

169 – selection  selections

175 – illness  illnesses

176 – self-guide  self-guided

187 – 189 – In this sentence what means “top three ones”?. I suggest: Travellers when asked about the most important insurances or to rate the insurances, 92% answered..

Discussion

216 – illness  illnesses

220 – from  of

222 – for  in

228-229 – I suggest you write it this way: The low percentage of travellers willing to seek pre-travel advice described in the literature contrasts with the percentage (percentage) of those motivated to buy travel health insurance in this study.

239 – 240 – You should clarify this sentence. According to what you described in 238-239, this sentence should be the cost of travel health insurance for the traveller or you are describing how much it cost without insurance?.

241 - 243 – I suggest: Overseas sickness coverage and medical evacuation are crucial while traveling during the Covid-19 pandemic since local healthcare capacity can be overwhelmed.

244 – remove “Besides” and “also”

245 – about  of

249 – of  to

255 – toward  of

256 – remove “is”

257 – one third  one-third

267 - 60% general  60% of general; give advice  advise; insurances  insurance

269 – the travelers  travelers; insurances  insurance

271 – insurances  insurance; a new  new

272 – airline  airlines

Conclusion

The conclusion should include something about knowledge related to infections, vaccines, and illnesses while traveling.

289 – 291 I suggest: between a third and half of the travelers understand the content of health insurance, almost all recognize its importance, and more than two-thirds were willing to buy a health insurance when traveling.

293 – 294 I suggest: Therefore, health professionals should be encouraged to acquire health insurance knowledge and transmitted it on during pre-travel consultation..

295 – consultation  consultations

Figure 1 must be redone, it is not possible to see due to lack of quality.

Reviewer #3: I would like to thanks the editors for giving me the opportunity to review the manuscript "Traveler’s Knowledge, Attitude, and Practice about Travel Health Insurance” which was submitted to ‘PLOS ONE'. I found the manuscript intriguing and informative. However, in my opinion, this manuscript cannot be published in its current form, certain corrections should be made. Therefore, based on my observations, I am making the following suggestions to improve the quality of the current work.

Comment one:

First and foremost, I'd like to thank to all of the authors for their strong effort. However, I found the manuscript's 'research gap and research objective' is not clear. The research gap should be clearly mentioned in the introduction part so that we can get a clear idea. For example, the following statements in the 'introduction section' is not clear and should be rewritten.

Introduction (lines no. 96 to 99):

'In addition, the travelers' awareness---------have not been well investigated'.

Comment two:

It is unclear Why the authors collected data on "willingness toward visiting travel medicine clinics". They emphasized on "travel insurance purchase behavior" rather than "willingness travel medicine clinics visit" in their research objectives. This should be included in your research objective or left out of the manuscript.

Questionnaire section (line no 119)

Comment three:

The authors mentioned about the pretest of the questionnaires, in the questionnaire section,which is commendable. However, the measurement scales of (lines 129 to 147) 'knowledge', 'attitude' and 'purchase behavior' are well established in the fields of marketing and management. It is unclear why the author did not adapted those scale to conduct the study. Moreover, a pretest alone is insufficient for developing a measurement scale, and it may cause concerns with the scale's validity and reliability. I believe that additional information in this area is required.

Comment four:

The findings of tables 1,2 and 3 should be properly discussed in the results section. The authors just mentioned a proportion of the table 1 and 2 and there was no interpretation for table 3. Please appropriately discuss the tables based on your research objective so that readers comprehend the research findings.

Moreover, no justification was provided for using one-way ANOVA and the t-test to analyse the data. Please provide justifications for using these statistical analysis to analyse the research data. Besides, the authors mentioned 'socio-demographic" variables in line 193 and 'socio-economic" variables inline 195 (under table 3). The terminology inconsistency should be addressed.

Comment five:

The data analysis from the discussion section can be transferred to the result section. It is better if the authors compare their findings to previous studies in this field and provide an explanation for any discrepancies in their findings based on prior research and theory. In the discussion section, they should also include the research contribution, which should address the research gap and objectives . Please revise the "Result" and "Discussion" sections.

Comment six:

It is mentioned that travelers rely more on "healthcare workers" (lines 241 to 246). However, the information about healthcare workers was not provided in any table.

Comment seven:

In the limitation section, the authors mentioned about non-response bias. This I believe is not a problem because 92.7% responses are already good enough. However, I am concerned that the the authors stated that they were unable to demonstrate a causal relationship due to the cross-sectional study (lines 254 to 256) !!!!! Cross-sectional studies are commonly used in social science studies to examine the causal relationship. There are numerous ways to addressing the bias of cross-sectional studies (e.g. marker variables). If the authors want, I believe they can use "Structural Equation Modeling" to provide greater understanding into the phenomenon. It is very important to understand the research objectives and what methodologies we are going to adopt to conduct the research.

Overall, I believe the manuscript can be accepted after addressing the above mentioned issues.

Thank you

Reviewer #4: The authors have addressed the reviewer's comments satisfactorily. An additional point should be mentioned in the discussion. How do the authors propose that travel medicine professionals educate themselves about the various types of travel insurance policies? This topic is rarely the exclusive topic of any conference session, although it is mentioned a lot in passing. Insurance medicine is itself a branch of medicine. Should there be greater cooperation between the two disciplines perhaps? Should we invite insurance physicians to attend our conferences or present at our webinars? A final point relates to the references. The authors correctly state that there has been limited previous work in this area. They may have overlooked the following article which is available as an open access source. There may be useful material that could strengthen your discussion. Source: Darrat M, Flaherty GT. An Exploratory Study of Medical Cover Policies Offered by the Travel Health Insurance Industry. International Journal of Medical Research & Health Sciences, 2019, 8(8): 1-8

7. PLOS authors have the option to publish the peer review history of their article (what does this mean?). If published, this will include your full peer review and any attached files.

Reviewer #1: No

Reviewer #2: No

Reviewer #3: No

Reviewer #4: No

---

## [Author Response · Author response to Decision Letter 1]

7 Nov 2022

Reviewer #2:

1. I suggest you add the study’s design in the title and in the abstract.

Response: Thanks for your recommendation. We have added the study’s design in the title, “Traveler’s Knowledge, Attitude, and Practice about Travel Health Insurance: A Community-based Questionnaire Study.” We also added the design in the abstract. “Methods: This was a cross-sectional study. Anonymous structured questionnaires were distributed to 1000 visitors to the Taiwan International Travel Fair in May 2019.”

2. In the abstract substitute:

Accident  accidents 

Response: We have revised the word. (LINE 34)

visitors of  visitors to the 

Response: We have revised the word. (LINE 40)

willingness  the willingness 

Response: We have revised the word. (LINE 45)

the international  international 

Response: We have revised the word. (LINE 36)

In the introduction substitute:

Increased  increasing

Response: We have revised the word. (LINE 70)

Tourisms  tourism 

Response: We have revised the word. (LINE 72)

Responsibilities  responsibility

Response: We have revised the word. (LINE 76)

Noncompliant  non-compliant

Response: We have revised the word. (LINE 77)

1980s’  1980s 

Response: We’ve removed this reference according to another reviewer’s suggestion. 

Shown  showed / has shown 

Response: We have revised the word. (LINE 87)

Categories  the categories 

Response: We’ve removed this sentence according to another reviewer’s suggestion. 

About  of

Response: Thanks for your recommendation. We’ve removed this sentence according to another reviewer’s suggestion. 

Risk-coverage  risk coverage

Response: We’ve removed this sentence according to another reviewer’s suggestion. 

a travel  travel

Response: We’ve revised the word. (LINE 94)

3. 77 - 80 – you should clarify this sentence. First you say it is the health providers responsibility and then you say travelers did not follow the suggestions. Maybe the responsibility is shared?

Response: We have revised the sentence to avoid the misunderstanding. (LINE 75-76)

4. 93 - 95 – How the first part of the sentence justifies the second part?

Response: Thanks for your question. Initially, we wanted to emphasize that although there were many kinds of travel health insurance, few studies investigated travelers’ knowledge, attitude, and practice. Thus, we conducted this study. To make it clearer, we removed the redundant sentences. (LINE 92-99)

5. In general, the introduction is not very clear and effective in demonstrating the importance of health insurance for travel and have several writing issues. I recommend that the text be proofread by a native English speaker.

Response: The introduction section is revised carefully. We have also sent it to the English editing by a native English speaker. 

6. 111 – 114 - You should clarify this paragraph. It is not clear what is the study design, the instrument and the participants.

Response: We’ve re-organized the paragraph of Design and Subjects to make them clearer. (LINE 104-112)

7. 117 – 119 – This phrase does not belong in the subjects’ section.

In the subject’ section you should add from where and how the participants were selected.

Response: We’ve re-organized the paragraph of Design and Subjects to make them clearer. (LINE 104-112)

8. Results

In the results’ section consider to include the number next to the percentage and also the p value next to the result presented.

The results presented in the tables should be better described in the text.

Response: Thanks for your recommendation. We’ve included the number and the p value next to the result presented. (Results paragraph) 

9. Grammar correction

169 – selection  selections

Response: We have revised the word. (LINE 162)

175 – illness  illnesses

Response: We have revised the word. (LINE 167)

176 – self-guide  self-guided

Response: We have revised the word. (LINE 169)

10. 187 – 189 – In this sentence what means “top three ones”? I suggest: Travelers when asked about the most important insurances or to rate the insurances, 92% answered.

Response: We appreciate your recommendation. Because another reviewer’s suggestion, we’ve adjusted this main finding to our discussion session. And we have revised the grammar according to your recommendation. (LINE 204-207)

11. Grammar correction

Discussion

216 – illness  illnesses

Response: We’ve removed this sentence according to another reviewer’s suggestion. 

220 – from  of

Response: We’ve removed this sentence according to another reviewer’s suggestion. 

222 – for  in

Response: We’ve removed this sentence according to another reviewer’s suggestion. 

12. 228-229 – I suggest you write it this way: The low percentage of travelers willing to seek pre-travel advice described in the literature contrasts with the percentage (percentage) of those motivated to buy travel health insurance in this study.

Response: We appreciate your recommendation. And we’ve revised the sentence according to your recommendation. (LINE 217-221)

13. 239 – 240 – You should clarify this sentence. According to what you described in 238-239, this sentence should be the cost of travel health insurance for the traveler or you are describing how much it cost without insurance?

Response: We apologize for confusing you. The cost is the emergency medical evacuation from a low-income nation. If the insurance coverage includes evacuation, then the traveler can pay less. We’ve revised this paragraph. We emphasized the respondents’ neglect of overseas sickness coverage and 24-hour emergency assistance first. Then, we showed that the fee of emergency medical evacuation is high. (LINE 231-234) 

14. 241 - 243 – I suggest: Overseas sickness coverage and medical evacuation are crucial while traveling during the Covid-19 pandemic since local healthcare capacity can be overwhelmed.

Response: Thanks for your suggestion. We’ve revised according to your recommendation. (LINE 238-239)

15. Grammar correction

244 – remove “Besides” and “also”

Response: We’ve removed this sentence according to another reviewer’s suggestion. 

245 – about  of

Response: We’ve removed this sentence according to another reviewer’s suggestion. 

249 – of  to 

Response: We’ve revised according to your recommendation. (LINE 240)

255 – toward  of

Response: We’ve removed this sentence according to another reviewer’s suggestion. 

256 – remove “is” 

Response: We’ve removed this sentence according to another reviewer’s suggestion.

257 – one third  one-third

Response: Thanks for your suggestion. We’ve revised the sentence to “Our study revealed only 31.5%–44.7% of respondents understood the content of travel insurance very well, whereas 9.5%–17.3% expressed their ignorance.” (LINE 222-223)

16. 267 - 60% general  60% of general; give advice  advise; insurances  insurance

Response: We’ve revised according to your recommendation. “ Recent studies have shown that approximately 60% of General Practitioners in New Zealand [7] and 39% of travel clinics worldwide [8] usually advise travelers to buy travel insurance.” (LINE 85-87)

17. 269 – the travelers  travelers; insurances  insurance

Response: We’ve revised the sentences after the last major revision. “Recent studies have shown that approximately 60% of General Practitioners in New Zealand [7] and 39% of travel clinics worldwide [8] usually advise travelers to buy travel insurance.” (LINE 85-87)

18. 271 – insurances  insurance; a new  new

Response: We’ve removed the sentences after the last major revision.

19. 272 – airline  airlines

Response: Thanks for your suggestion. We’ve removed the sentences after the last major revision. 

20. Conclusion

The conclusion should include something about knowledge related to infections, vaccines, and illnesses while traveling.

Response: We’ve added the sentence, “Understanding local epidemic and implement personal protective measures is still important.” in Conclusion paragraph. (LINE 272-273)

21. 289 – 291 I suggest: between a third and half of the travelers understand the content of health insurance, almost all recognize its importance, and more than two-thirds were willing to buy a health insurance when traveling.

Response: Thanks for your suggestion. We’ve revised the sentence according to your recommendation. (LINE 269-271)

22. 293 – 294 I suggest: Therefore, health professionals should be encouraged to acquire health insurance knowledge and transmitted it on during pre-travel consultation.

Response: We’ve revised the sentence according to your recommendation. (LINE 274-276)

23. 295 – consultation  consultations 

Response: We’ve revised the word according to your recommendation. (LINE 276)

24. Figure 1 must be redone, it is not possible to see due to lack of quality.

Response: Thanks for your suggestion. We’ve removed Figure 1 according to the last reviewer’s suggestion. 

Reviewer #3:

I would like to thank the editors for giving me the opportunity to review the manuscript "Traveler’s Knowledge, Attitude, and Practice about Travel Health Insurance” which was submitted to ‘PLOS ONE'. I found the manuscript intriguing and informative. However, in my opinion, this manuscript cannot be published in its current form, certain corrections should be made. Therefore, based on my observations, I am making the following suggestions to improve the quality of the current work.

1. First and foremost, I'd like to thank to all of the authors for their strong effort. However, I found the manuscript's 'research gap and research objective' is not clear. The research gap should be clearly mentioned in the introduction part so that we can get a clear idea. For example, the following statements in the 'introduction section' is not clear and should be rewritten.

Introduction (lines no. 96 to 99):

'In addition, the travelers' awareness---------have not been well investigated'.

Response: Thanks for your comments. We’ve revised the paragraph of introduction according to your suggestion. (LINE 92-99)

2. It is unclear Why the authors collected data on "willingness toward visiting travel medicine clinics". They emphasized on "travel insurance purchase behavior" rather than "willingness travel medicine clinics visit" in their research objectives. This should be included in your research objective or left out of the manuscript.

Response: Thanks for your comments. Pretravel consultation is also important in addition to buying travel health insurance. In fact, pretravel consultation is our prime opportunity to educate travelers. Thus, we want to understand the correlation to clarify what else we could do. So, we include the willingness to seek pre-travel consultation in our research objective. (LINE 94-95)

Questionnaire section (line no 119)

3. The authors mentioned about the pretest of the questionnaires, in the questionnaire section, which is commendable. However, the measurement scales of (lines 129 to 147) 'knowledge', 'attitude' and 'purchase behavior' are well established in the fields of marketing and management. It is unclear why the author did not adapt those scale to conduct the study. Moreover, a pretest alone is insufficient for developing a measurement scale, and it may cause concerns with the scale's validity and reliability. I believe that additional information in this area is required.

Response: Thanks for your comments. Reviewing the literatures, most studies focus on health insurance rather than travel health insurance [1-3]. Our last study focused on emergency medical assistance services [4]. Because this time we wanted to know travelers’ knowledge, attitude, and purchase behavior of different kinds of travel insurance, we used our questionnaire which designed by ourselves. The Cronbach’s Alpha is 0.927, which represents good reliability. About validity, ten physicians with clinical experienced in travel medicine conducted face validity. So, we believe our questionnaire is qualified. 

Reference:

1. Sheryl AR, Margarete E, Charles EI. Adolescents’ knowledge of their health insurance coverage. J Adolesc Health. 1998;22:293-299 

2. Dhungana BR, Giri SS, Kushwaha SP, Khanal DK, Yadav BK. Awareness and Perception Regarding Health Insurance Among Community People in a Municipality: A Cross Sectional Study. MedS J Med Sci. 2021;1:76-82.

3. Evita A, Enrico MB, Gian PS. Understanding Insurance Knowledge: A Brief 7-Item Measure. Int J Bus Manag. 2021,16 (2), 65-74.

4. Lee YH, Lu CW, Wu PZ, Huang HL, Wu YC, Huang KC. Attitudes and awareness of medical assistance while traveling abroad. Global Health. 2018;14(1):67. doi:10.1186/s12992-018-0382-5.

4.The findings of tables 1,2 and 3 should be properly discussed in the results section. The authors just mentioned a proportion of the table 1 and 2 and there was no interpretation for table 3. Please appropriately discuss the tables based on your research objective so that readers comprehend the research findings.

Response: Thanks for your comments. Table 2 is our main research objective, so we wrote more about table 2. We’ve revised the discussion paragraph according to your suggestion. Our interpretation for table 3 is on LINE 240-246. We emphasize travelers who have the willingness to buy travel health insurance may not have good knowledge of travel medicine.

5. No justification was provided for using one-way ANOVA and the t-test to analyze the data. Please provide justifications for using these statistical analyses to analyze the research data. Besides, the authors mentioned 'socio-demographic" variables in line 193 and 'socio-economic" variables inline 195 (under table 3). The terminology inconsistency should be addressed.

Response: Thanks for your recommendation. We’ve revised the terminology inconsistency. There were 10 questions to test travelers’ knowledge of travel medicine. Each question was scored 1 point for a correct response, with a total score of 10 points, so it was a continuous variable. The socio-demographic variables included age, medical history, travel-associated illness and planned special activities during travel. We divided the respondents into 2 and 5 groups, so we used one-way analysis of variance (5 groups) and independent t-test (2) groups to clarify the relation between the knowledge of travel medicine and socio-demographic variables. 

6. The data analysis from the discussion section can be transferred to the result section. It is better if the authors compare their findings to previous studies in this field and provide an explanation for any discrepancies in their findings based on prior research and theory. In the discussion section, they should also include the research contribution, which should address the research gap and objectives. Please revise the "Result" and "Discussion" sections.

Response: Thanks for your recommendation. We’ve revised the Result and Discussion sections according to your suggestion. 

6. It is mentioned that travelers rely more on "healthcare workers" (lines 241 to 246). However, the information about healthcare workers was not provided in any table.

Response: We’ve added Table 4 to show the source and subjective trust of travel health insurance information. 

7. In the limitation section, the authors mentioned about non-response bias. This I believe is not a problem because 92.7% responses are already good enough. However, I am concerned that the authors stated that they were unable to demonstrate a causal relationship due to the cross-sectional study (lines 254 to 256) ! Cross-sectional studies are commonly used in social science studies to examine the causal relationship. There are numerous ways to addressing the bias of cross-sectional studies (e.g. marker variables). If the authors want, I believe they can use "Structural Equation Modeling" to provide greater understanding into the phenomenon. It is very important to understand the research objectives and what methodologies we are going to adopt to conduct the research.

Response: Thanks for your recommendation. We agreed! We’ve revised the Limitations section. This is the pilot study to understand the traveler’s knowledge, attitude, and practice about travel health insurance. We’ll analyze the factors which influence their behavior in the future. 

Reviewer #4:

 The authors have addressed the reviewer's comments satisfactorily. An additional point should be mentioned in the discussion. How do the authors propose that travel medicine professionals educate themselves about the various types of travel insurance policies? This topic is rarely the exclusive topic of any conference session, although it is mentioned a lot in passing. Insurance medicine is itself a branch of medicine. Should there be greater cooperation between the two disciplines perhaps? Should we invite insurance physicians to attend our conferences or present at our webinars? A final point relates to the references. The authors correctly state that there has been limited previous work in this area. They may have overlooked the following article which is available as an open access source. There may be useful material that could strengthen your discussion. Source: Darrat M, Flaherty GT. An Exploratory Study of Medical Cover Policies Offered by the Travel Health Insurance Industry. International Journal of Medical Research & Health Sciences, 2019, 8(8): 1-8

Response: We appreciate your recommendation. If there are more cooperation between the two disciplines, we believe that travelers can receive more comprehensive care. (LINE 219-221) We have also added this study finding into our discussion section. (LINE 224-226, 254-255) (Darrat M, Flaherty GT. An Exploratory Study of Medical Cover Policies Offered by the Travel Health Insurance Industry. International Journal of Medical Research & Health Sciences, 2019, 8(8): 1-8)

---

## [Decision Letter · Decision Letter 2]

18 Jan 2023

Traveler's Knowledge, Attitude, and Practice about Travel Health Insurance: A Community-based Questionnaire Study.

PONE-D-21-34714R2

Dear Dr. Huang,

We’re pleased to inform you that your manuscript has been judged scientifically suitable for publication and will be formally accepted for publication once it meets all outstanding technical requirements.

Kind regards,

Andrea Cioffi

Academic Editor

PLOS ONE

Additional Editor Comments (optional):

Reviewers' comments:

Reviewer's Responses to Questions

**Comments to the Author**

1. If the authors have adequately addressed your comments raised in a previous round of review and you feel that this manuscript is now acceptable for publication, you may indicate that here to bypass the “Comments to the Author” section, enter your conflict of interest statement in the “Confidential to Editor” section, and submit your "Accept" recommendation.

Reviewer #1: All comments have been addressed

Reviewer #4: All comments have been addressed

2. Is the manuscript technically sound, and do the data support the conclusions?

Reviewer #1: Yes

Reviewer #4: Yes

3. Has the statistical analysis been performed appropriately and rigorously? 

Reviewer #1: Yes

Reviewer #4: Yes

4. Have the authors made all data underlying the findings in their manuscript fully available?

Reviewer #1: Yes

Reviewer #4: Yes

5. Is the manuscript presented in an intelligible fashion and written in standard English?

Reviewer #1: Yes

Reviewer #4: Yes

6. Review Comments to the Author

Reviewer #1: N/A — all comments addressed to my satisfaction and I have nothing more to add.

Thank you for having me review.

Reviewer #4: Thank you for revising your manuscript in line with reviewer recommendations. All points have been addressed.

7. PLOS authors have the option to publish the peer review history of their article (what does this mean?). If published, this will include your full peer review and any attached files.

Reviewer #1: No

Reviewer #4: No

---

## [Editor Report · Acceptance letter]

31 Jan 2023

PONE-D-21-34714R2 

Traveler’s Knowledge, Attitude, and Practice about Travel Health Insurance: A Community-based Questionnaire Study. 

Dear Dr. Huang:

I'm pleased to inform you that your manuscript has been deemed suitable for publication in PLOS ONE. Congratulations! Your manuscript is now with our production department. 

Kind regards, 

on behalf of

Dr. Andrea Cioffi 

Academic Editor

PLOS ONE